# Pharmaceutical Residues in Senior Residences Wastewaters: High Loads, Emerging Risks

**DOI:** 10.3390/molecules26165047

**Published:** 2021-08-20

**Authors:** Silvia Lacorte, Cristian Gómez-Canela, Carole Calas-Blanchard

**Affiliations:** 1Department of Environmental Chemistry, IDAEA-CSIC, Jordi Girona 18-26, 08034 Barcelona, Catalonia, Spain; 2Department of Analytical Chemistry and Applied (Chromatography Section), School of Engineering, Institut Químic de Sarrià-Universitat Ramon Llull, Via Augusta 390, 08017 Barcelona, Spain; cristian.gomez@iqs.url.edu; 3Biocapteurs-Analyses-Environnement, Université de Perpignan Via Domitia, 52 Av. Paul Alduy, CEDEX, 66860 Perpignan, France; carole.blanchard@univ-perp.fr; 4Laboratoire de Biodiversité et Biotechnologies Microbiennes, USR 3579 Sorbonne Universités (UPMC) Paris 6 et CNRS Observatoire Océanologique, 66650 Banyuls-sur-Mer, France

**Keywords:** senior residences, wastewater, pharmaceuticals, sewage grid

## Abstract

Senior residences are health-care facilities that are socially-accepted for the assistance of elderly people. Since the elderly account for the foremost pharmaceutical-consuming age-group, senior residences become a hot-spot for pharmaceuticals discharge to the sewage grid. The objectives of the present study were to identify the bioactive pharmaceuticals in sewage waters from senior residences and to propose an on-site monitoring strategy for their control. In this study, we have studied the presence of 43 pharmaceuticals highly consumed by the elderly population in six senior residences located in Spain, France and Portugal. Wastewater was sampled directly from the water-chest in each residence during different times of the day throughout one week. Main compounds detected at the high µg L^−1^ level were analgesic and antipyretic drugs such as acetylsalicylic acid, paracetamol, ibuprofen; antibiotics such as amoxicillin and sulfamethoxazole; compounds for the treatment of neuropathies as gabapentin, trazodone and valsartan; pharmaceuticals for the treatment of diabetes (vildagliptin) and anticancer drugs. The daily loads discharged were estimated and their fate was evaluated. The final objective of this study is to highlight the need to implement at-source waste water treatment procedures in senior residences, which have been identified as a point source pollution of pharmaceuticals.

## 1. Introduction

More than 20 years have passed since pharmaceuticals have been detected in river waters as a result of their extensive use in human treatments and veterinary purposes [1,2]. Pharmaceuticals account for a total pharmaceutical market value of 220,200 M€ in Europe, with more than 4000 active ingredients and production volumes of tons [3]. After administration, many drugs are excreted unchanged or as metabolites by urine and feces, and reach the aquatic systems either via Wastewater Treatment Plants (WWTPs) [4,5], wetlands [6], untreated effluents or run-off [7].

The overwhelming problem of pharmaceutical pollution is especially relevant in urban areas suffering from water scarcity [8]. This type of pollution is exacerbated due to the increased consumption of drugs as the population ages. According to the United Nations, in Europe, Japan, North America and some regions of China the phenomenon of aging and over-aging has led to societies where 14–23% of the population is over 65, and this percentage is expected to grow in the next 10 years [9]. This over-aging phenomenon is reaching concerning levels in Portugal, Italy, Germany, Finland and Greece which account for the worst percentages in the UE. In plain figures, the number of people aged above 65 years is tens of millions and varies by country. This age-group consumes a higher quantity of pharmaceuticals than the rest of the population, with an estimated intake 5 to 10 pills inhab^−1^ day^−1^ [10]. The most consumed pharmaceuticals are within the category of nervous system (N, according to the Anatomic Therapeutic Code, ATC), alimentary tract (A), anti-infective (J) and antineoplastic for the treatment of cancer (L) and the doses administered increase compared to the younger population [11].

In the last years the homes for the elderly and socio-sanitary centers have become a good option to provide services, an adequate milieu and ensure the public health according to the demand and needs of the elderly people. The homes for the elderly are infrastructures that articulate diverse services in response to biopsychosocial needs and have become popular in most European countries. Senior residences have a configuration of typically 50–150 individuals and provide lodging, meal services and health assistance. Following medical prescription patterns, the total amount of pharmaceuticals consumed in conventional senior residences range from 1 to 400 g day^−1^, and concomitantly hundreds of milligrams of drugs of different therapeutic use are excreted [12]. Wastewaters from senior residences, as in most hospitals, are discharged untreated to the sewage grid and enter into the WWTPs. As many pharmaceuticals are not eliminated with the conventional activated sludge treatment, these compounds are discharged to receiving waters, posing environmental risks [5]. Considering that senior residences have become a popular and well-accepted option for the health-care system, especially in southern Europe due mainly to the mild climate, these establishments have become a hot-spot for pharmaceutical discharge [12].

The objective of the present study was to monitor the presence of 43 pharmaceuticals in the effluents of six senior residences, located in Spain, France and Portugal (Table 1) in order to determine the type and concentration of widely consumed pharmaceuticals and the hourly/daily variations. Based on compounds detected, the total discharge in the sewage grid has been estimated according to their ATC group and their fate evaluated. Finally, pharmacological data of studied drugs is provided as well as a list of main metabolites excreted that can be future compounds to be analyzed in water. Studied target compounds (Table 2) were selected based on it their consumption by the elderly according to previous published papers about the Predicted Environmental Concentrations (PECs) calculated in senior residences [12] and regionally [13]. Therefore, this is one of the first papers focused on the determination of pharmaceuticals directly in the effluents from senior residences.

## 2. Results and Discussion

### 2.1. Pharmaceuticals Released from Senior Residences

The total concentration of pharmaceuticals in the effluents collected directly in the 6 senior residences is indicated in Figure 1 according to the sampling hour (9 h, 13 h, 16 h) and during one week. Among the different senior residences studied, SP1 accounted for the highest concentration and this was expected as it corresponds to a socio-sanitary center where the degree of illness is much higher than in senior residences and thus the amount of pharmaceuticals administered [12]. Considering the 5 other senior residences, the mean levels decreased in the order SP2 > FR1 > PT1 > PT2 >> FR2. Considering all sampling points in each residence, the total concentrations ordered from the highest to lowest in each country ranged from 72 to 4209 µg L^−1^ in SP1, 219 to 1800 µg L^−1^ in SP2, 109 to 2292 µg L^−1^ in FR 1, 18 to 52 µg L^−1^ in FR 2, 14 to 2352 µg L^−1^ in PT1 and 4 to 326 µg L^−1^ in PT 2. These very high concentrations at the µg L^−1^ level are explained by the fact that samples corresponded to the morning and afternoon flush waters and were collected from the chest without any dilution with water from laundry or showers (in general showers are scheduled latter), and correspond to the pharmaceuticals excreted through urine and feces by the residents not yet diluted from the general sewage grid. Considering the mean concentration of all drugs detected in each residence, specific contamination patterns were observed. The Pearson correlation coefficient (r) was of 0.884 between SP 1 and SP2 and indicated a high relationship among the pharmaceuticals detected in these 2 residences. FR 1 was well correlated with PT 1 and PT 2 with r of 0.872 and 0.985 and PT 1 and PT 2 had an r of 0.919. No other relationships were found.

Within each residence, no correlation was observed among sampling days except for SP 1 in 16 and 17 May (*r* = 0.884), SP 2 in 18 and 19 May (*r* = 0.862) and PT 2 in 27 and 30 June (*r* = 0.986). We assessed samples from 3 different periods of the day (except for SP 2 and FR 2 and Portugal which were sampled only in the morning and in the afternoon) to cover the daily variability. We found differences in the detected concentrations according to the sampling period suggesting that the punctual sampling provides a “picture” of the precise moment that bathrooms are flushing (Figure 1). The sampling at 9 h corresponds to the time residents are waking-up and going to the bathroom, and in here we observed that the concentrations were slightly higher than in other sampling periods, especially in residences from Spain. At 13 h, it is the time before lunch that the bathroom is used and at 16–17 h corresponds to the time after the lunch rest where the bathroom is used again. We avoided sampling after 10 h as it is the time of washing machines and showers which would dilute the pharmaceuticals excreted. The intra-day and inter-day variability was high in SP although there was a good correlation among samples collected at 9, 13 and 16 h (*r* > 0.645), with some peak concentrations in the morning sample attributed to specific high concentration on some of the drugs analyzed which might be explained by the coincidence of collecting water at the time these pharmaceuticals were excreted. In FR, the concentration of the different drugs did not vary so much and this was recognized by a higher correlation among days (*r* of 0.716, 0.720 and 0.895 in 3 days). Specifically, in FR 2 the overall concentrations were much lower and more constant considering daily variations (*r* from 0.646 to 0.936 for intra-day variability and *r* between 0.616 and 0.989 for inter-day variability at 9 and 16 h), indicating a very similar contamination pattern among days. In Portugal, no correlation was observed by sampling in different hours. The great differences of total pharmaceuticals detected in the different days (and hours) largely vary because grab sampling is only able to detect those contaminants in the punctual period that sampling has been performed, so there is high variability depending on “the chance” to collect the flushes with the highest levels of pharmaceuticals. However, due to sampling constraints or unavailability of personnel from the residences to assist in the sampling, not all days and times could be sampled in each residence and this makes comparability and statistical analysis difficult.

The minimum, maximum and 25th and 75th quartile concentrations detected in the 6 senior residences are indicated in Figure 2. Compounds displayed were present in more than 50% of the samples and are ordered from the highest to the lowest 75th quartile concentrations. The number of pharmaceuticals detected in each residence varied among sampling days and ranged between 20 and 30 in SP 1 and SP 2, from 16 to 23 in FR 1, from 13 to 17 in FR 2, and 18–20 in PT 1 and 9–22 in PT 2. The high detectability of contaminants reflects the compounds consumed by the elderly. Appendix A of the Supplementary Information indicate the concentrations detected in each residence.

Pharmaceuticals detected in all senior residences were: L-ascorbic acid, macrogol, acetylsalicylic acid, furosemide, carbamazepine, paracetamol and diclofenac. All these compounds were preferentially prescribed in the studied residences [12]. Caffeine was also detected as being mostly attributed to coffee, tea, refreshments or products containing cocoa or chocolate, but also as a component of cold medicines, painkillers, and stimulants. The highest concentrations of caffeine were found in SP 1, SP 2 and FR 2 (Figure 2) and surprisingly were within those reported in environmental waters worldwide [15], indicating that senior residences do not represent an extra source of caffeine compared to the general population. Caffeine is a marker of urban contamination but also has environmental impacts [16], indicating that its monitoring is well-justified. Compounds detected in 5 residences (among the 6) in a reiterative way although with varying levels were valsartan, cyclophosphamide, ibuprofen, donepezil, levetiracetam, dichlorobenzyl alcohol and amylmetacresol. For other compounds, we observed a residence specific pattern which is in line with the different prescription doses which mainly vary among countries. For instance, estrone was only detected in FR 1 and FR 2 and this is in accordance to the standardized use of this compound for perimenopausal symptoms in France but not in Spain or Portugal (personal communication). On the other hand, megestrol was only detected in Spanish residences. Surprisingly also, gabapentin, quetiapine and trazodone, all of them belonging to the N (Nervous system) category, were only detected in Spain and Portugal and this might be also related to prescription models. Ifosfamide was detected in all residences from France and Portugal but never detected in Spain. L-ascorbic acid was detected at much higher concentrations in residences from France and Portugal than from Spain. Other compounds shown in Figure 2 were also residence-specific. Overall, these detected pharmaceuticals are in general excreted in relatively high percentages (Table 2) and are detected in river waters on a regular basis [17,18].

Compounds detected sporadically (less than 50% of the samples) were bicalutamide and dextromethorphan only detected in SP1 and tiotropium was occasionally detected. Chlormethiazole, chlorpheniramine and clarithromycin were only detected in SP 1 and SP 2, with the highest concentrations in SP 1, and cloperastine in SP 2 at very low concentrations (Appendix A). Dutasteride was the only compound not detected in any sample as the dose administered is low (0.5 mg day^−1^) and it has low excretion rates (Table 2) as it undergoes extensive hepatic metabolism mediated by CYP3A4 and CYP3A5 and is excreted as metabolites. Therefore it is unlikely that it is detected in waters.

### 2.2. Main Patterns and Potential Metabolites

Figure 3 shows the main patterns of pharmaceuticals detected in each residence. In all residences, the most consumed were pharmaceuticals for the treatment of the nervous system (ATC code N), followed by drugs for the treatment of cardiovascular diseases (ATC code C) and antiinfectives (ATC code J). Although in minor proportions, drugs from other ATC codes were also identified. Because the concentrations detected are in general high, and some of the drugs can be excreted as metabolites, the identification of metabolites gain importance. In this study we made a theoretical approach on potential metabolites that could be potentially released from wastewaters from senior residences detected. Table 2 indicates the pharmacological information extracted from the Drugbank database [14] that is used to define whether drugs are excreted in a larger or minor proportion and serves to identify compounds that will be detected in receiving waters as unchanged drug or as metabolites. The levels detected in wastewaters depend also on the rate at which drugs are eliminated from the body and compounds with very small half-life will be rapidly excreted (Table 2). Some compounds as dichlorobenzyl alcohol, amoxicillin, acetylsalicylic acid, carbamazepine, diclofenac, ezetimibe, fluticasone, gabapentin, levetiracetam, levofloxacin, macrogol, pregabalin, rosuvastatin, tiotropium and valsartan are excreted unchanged at >60% of the administered dose. This means that their chances to be detected are very high. In fact, these are the most ubiquitous compounds detected in wastewaters from senior residences. Our empirical data coincides with previous studies indicating that antipsychotics, laxatives, benzodiazepines and antiplatelets among others are highly consumed by the elderly [11,19] and have been detected in river waters with quite high frequency [8,17,18,20].

Upon administration, pharmaceuticals can be metabolized in the liver and in a lesser extent in the kidney, hydrolyzed in plasma or can be excreted unchanged. Table 2 also provides a list of metabolites identified according to the Drugbank database [14]. These metabolites are mainly the hydroxylated, decarboxylated or oxydated forms of the parental drug and also glucuronide or sulfate conjugates. The presence of metabolites in urine is a first step to track their identification in river waters, as recently reported [21,22,23]. Because some metabolites are the active form of pharmaceuticals, it is envisaged to include them in future monitoring programs.

### 2.3. Total Discharges and Fate

Figure 4 shows the amount of pharmaceuticals discharged to the sewage grid on a daily basis. Values ranged from 0.4 to 17 g day^−1^ for residences of 52 to 130 beds, being the Spanish residences the ones discharging the highest amounts of contaminants, in line with their higher levels detected compared to the rest, also considering that they are bigger in size. The main categories of pharmaceuticals are also indicated, showing that neurological drugs (N) are the most consumed, followed by cardiovascular (C), alimentary tract (A), neoplastic (L), respiratory (R) or anti-infective (J), depending on the residence. This is of relevance for neurotoxic compounds as they can have negative effects on biota [24], antibiotics can be absorbed by plants upon irrigation with river or reclaimed water [25] or anticancer drugs can have chronic effects on aquatic biota [26].

Considering that senior residences have become very popular in southern countries, the daily amounts of pharmaceuticals discharged to the sewage grid can be relevant. Considering Catalonia, with a population of 7.5 M inhabitants, the number of residences is of 60.954 [27], with almost one third in Barcelona serving a population of 3 M persons. Based on these figures, the total amount of pharmaceuticals discharged are rapidly rocketing. Considering a grid dilution factor of 1000, one order of magnitude higher that the value proposed in a previous paper [12], each residence discharges 17 g of active ingredients per day (considering e.g., SP 1) to the sewage grid, which multiplied by the number of residences in Catalonia represents kg of pharmaceuticals entering the different WWTPs. A similar situation would be found for hospitals [28], but despite their higher capacity, the number of hospitals in a city is much smaller. Therefore, effluents from health institutions can be identified as an important and continuous source of pharmaceuticals to the sewage-grid, and thus, to receiving waters considering that WWTP operating with conventional treatment do not have the capacity to eliminate the total loads of pharmaceuticals received [4,5]. Despite the elimination of pharmaceuticals during WWTP and the dilution factor once discharged to receiving waters, many compounds are detected at concentrations from 0.001 to 1 µg L^−1^, and in some cases exceeding the Environmental Quality Standards [17]. According to the Pharmaceutical Strategy of the EU [29], there is an imminent need to reduce wastage and improve the management of waste containing pharmaceutical residues. In this direction, actions are required to reduce the release of these compounds using new technological solutions for at-source treatment as recently proposed [30]. Other initiatives include social, governmental, medical and analytical actuations to minimize the presence of pharmaceuticals in the environment [31] so that risk perception regarding pharmaceutical contamination can be contained [32].

## 3. Material and Methods

### 3.1. Chemicals and Reagents

Thirty-three standards of 98–99% purity were purchased from Sigma-Aldrich (St. Louis, MO, USA) and ten from Santa Cruz Biotechnology (Santa Cruz, CA, USA). Acetaminophen-methyl-d_3_ (Sigma-Aldrich) was used as internal standard. All the target compounds and the main uses and category according to the ATC code are indicated in Table 2. Stock standard solutions were prepared at a concentration of 1000 ng µL^−1^ and working solutions at 1–10 ng µL^−1^, all in methanol.

Methanol (MeOH) and HPLC water (LiChrosolv grade) were supplied by Merck (Darmstadt, Germany). Formic acid (HCOOH), hydrochloric acid 37% (HCl), and ammonium acetate (NH_4_OAc) were supplied by Sigma-Aldrich (St. Louis, MO, USA) and acetonitrile (ACN) was supplied by Fisher Scientific Chemical (Bridgewater, NJ, USA).

### 3.2. Study Site and Sample Collection

Samples were collected from 6 senior residences located in north-east Spain (SP 1 and SP 2), in south of France (FR 1 and FR 2), and Portugal (PT 1 and PT 2). The characteristics of each residence, number of residents, the number of pharmaceuticals administered and water consumption are indicted in Table 1. These residences host general medicine patients except in S1 which is a socio-sanitary center administering also drugs for psychiatric treatment, neurologic, palliative assistance and traumatology.

In all cases, samples were collected from the wastewater chest located inside the residences’ premises. These chests collect all the water used within each residence including flush, tap water and laundry, and are thereafter, connected with the municipal sewage grid. The number of samples collected in each residence are listed in Table 1 and varied depending on the accessibility in each residence and availability of maintenance personnel that was essential to obtain water from the chests. To avoid dilution from laundry-waters, samples were collected in different times according to the advices given by personnel from the residences. In general, samples were collected 2 or 3 times between 9 and 17 h to have intra-day variability and during one week (5 days) to have inter-day variability, except in SP 1 and FR 2 that in some days only 2 samples were collected and in Portugal (PT 1 and PT 2) that only 2 samples were collected in the morning and at 16 h in 4 days instead of 5. Wastewater was collected using a telescopic rod connected to a glass-jar and were dosed in an amber glass bottles. To obtain 1 L of water this operation was repeated 5–7 times. Samples were transported refrigerated to the main laboratory in IDAEA-CSIC and upon reception were centrifuged at 3000 rpm for 10 min and then filtered through 0.45 µm nylon membrane filters (Phenomenex, Torrance, CA, USA) to remove the large amount of solid material and particulate matter. Samples were stored at 4 °C and extracted within 2–3 days to avoid degradation of pharmaceuticals. To ensure representative sampling, identical sampling strategies were used for all the residences. Sampling was performed May 2017 for Barcelona’s residences, June 2017 for Portuguese residences and July 2017 for French residences.

### 3.3. Sample Extraction and Analysis

Samples were extracted using solid phase extraction (SPE) according to a previous study [19] where the extraction and the mass-spectrometric conditions were optimized to have high extraction yields and low errors for a suite of pharmaceuticals consumed by the elderly. Briefly, samples were acidified with HCl 0.1 N at pH 2, and spiked with the internal standard acetaminophen-methyl-d_3_ at 0.1 µg L^−1^. One hundred mL of filtered water was extracted using Oasis HLB cartridges (200 mg, Waters, Milford, MA, USA), rinsed with 2 mL of Milli-Q water to eliminate potential interferences and eluted with 6 mL of HCOOH:MeOH (5:95) and 6 mL of MeOH. Extracts were then evaporated to almost dryness in a TurboVap^®^ LV (Uppsala, Sweden) under a stream of N_2_, filtered with a 0.22 µm polytetrafluoroethylene (PTFE) syringe filter (Advantec, Tokyo, Japan) and transferred to a 2 mL chromatographic vial, evaporated to dryness with a ReactiVap^®^, and reconstituted with 200 µL of a 30:70 (*v*/*v*) ACN:HPLC water mixture. Samples were analyzed by liquid chromatography coupled to a triple quadrupole detector (Xevo TQS, Acquity Waters, Milford, CT, USA) (LC-MS/MS) with the conditions optimized in a previous study [14]. A Synergy Polar-RP column (250 mm × 4.6 mm, particle size 4 µm, Phenomenex, Torrace, USA) was used with the mobile phase consisting in (A) 0.1% of formic acid in acetonitrile and (B) 0.1% formic acid in water. Gradient elution started at 10%A, increasing to 70% A in 20 min and to 100% of A in 5 min, held for 7 min and returned to initial conditions after 3 min. The flow rate was set at 0.4 mL min^−1^ and the volume injection was 10 µL. Except for aspirin and furosemide that were detected in negative electrospray ionization (ESI−), the rest of the compounds were measured under positive electrospray ionization (ESI+). Mode of acquisition was selected reaction monitoring (SRM) using two transitions from the precursor ion to the product ion to identify each pharmaceutical. Data was processed using the MassLynx software (v4.1, Waters, Milford, MA, USA). Internal standard quantification was performed using acetaminophen-methyl-d_3_. Calibration was performed over a concentration range from 0.001 to 2.5 ng µL^−1^. Quality control parameters for wastewaters are indicated in a previous paper on method development [19]. Method detection Limits (MDL) calculated with wastewater spiked at a concentration of 0.1 µg L^−1^ are indicated in Appendix A of the Supplementary information.

## 4. Conclusions

This study shows that senior residences are a point source for pharmaceutical discharge to the sewage grid, and hence, to receiving waters. The analysis of 43 drugs consumed by the elderly has revealed that effluents from senior residences contained from 14 to 30 drugs at concentrations at the high µg L^−1^ level, and that in most cases the concentrations varied according to the sampling period. Some compounds such as L-ascorbic acid, macrogol, acetylsalicylic acid, furosemide, carbamazepine, paracetamol and diclofenac were systematically detected in all senior residences and corresponded to those compounds that are excreted unchanged in large proportions. Other compounds were residence-specific and corresponded most probably to the medical prescriptions in each residence or country. This study also highlights the importance to use pharmacological data (e.g., Drugbank database) to determine the extent of excretion of the parental drug and the metabolites formed, as they can be potentially identified in waters. According to the concentration of pharmaceuticals in each residence and the water use, the total amount of drugs discharged to the sewage grid varied from 0.4 g to 17 g day^−1^, indicating that senior residences are important sources of pharmaceutical pollution and that treatment at the source is essential for minimizing their subsequent release to the environment.

## Figures and Tables

**Figure 1 molecules-26-05047-f001:**
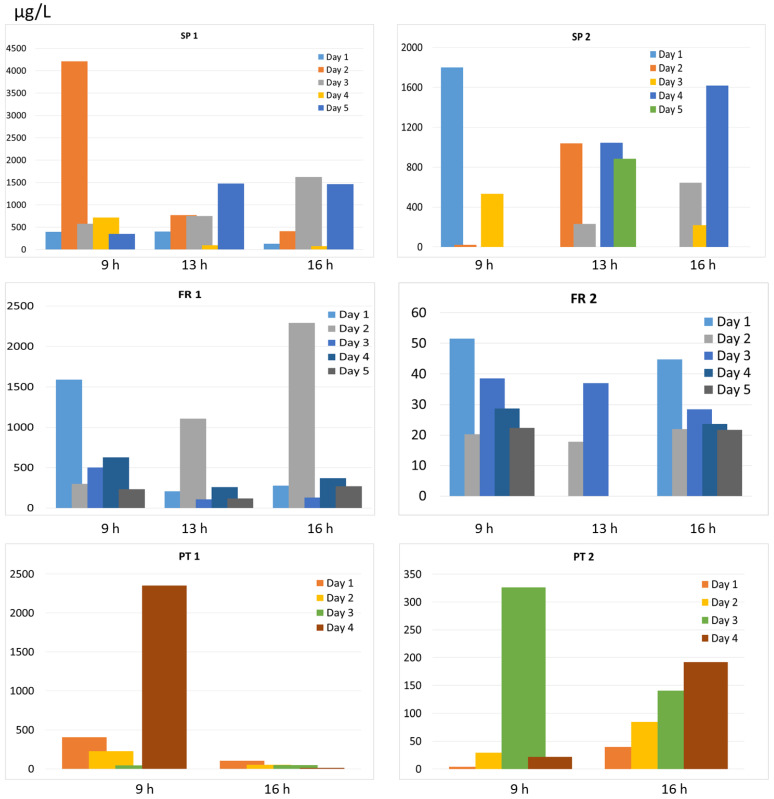
Total pharmaceuticals detected in water (µg L^−1^) in each sampling day within one week and at the different time frames (morning and afternoon flushes) in senior residences from Spain (SP 1 and SP 2), France (FR 1 and FR 2) and Portugal (PT 1 and PT 2).

**Figure 2 molecules-26-05047-f002:**
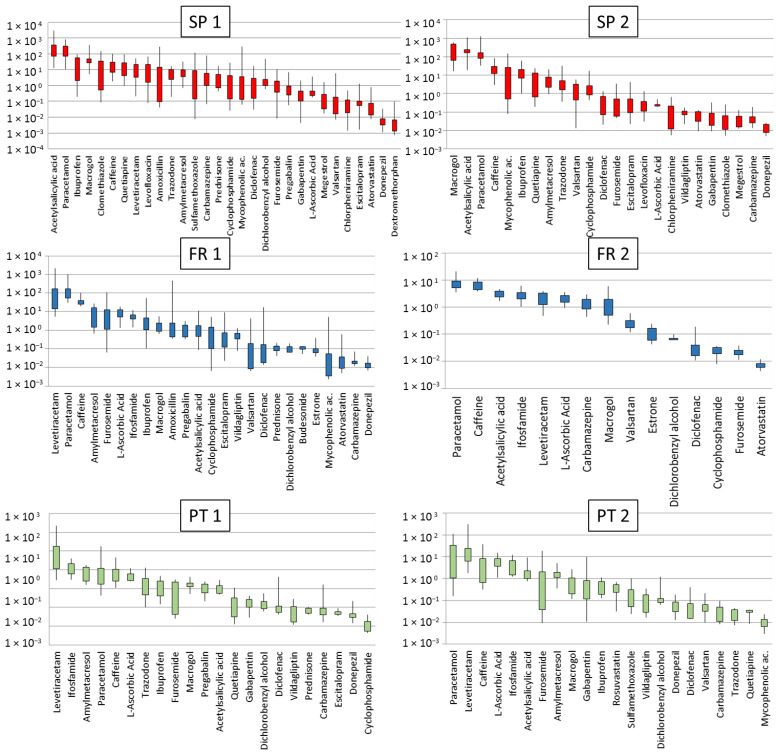
Boxplots indicating the minimum, maximum, 25th and 75th percentile concentrations of the different drugs detected (µg L^−1^) in more than 50% of the samples in each senior residence, ordered from the highest to lowest concentrations (red SP, blue FR and green PT). The Y axis is in logarithmic scale to allow better comparison.

**Figure 3 molecules-26-05047-f003:**
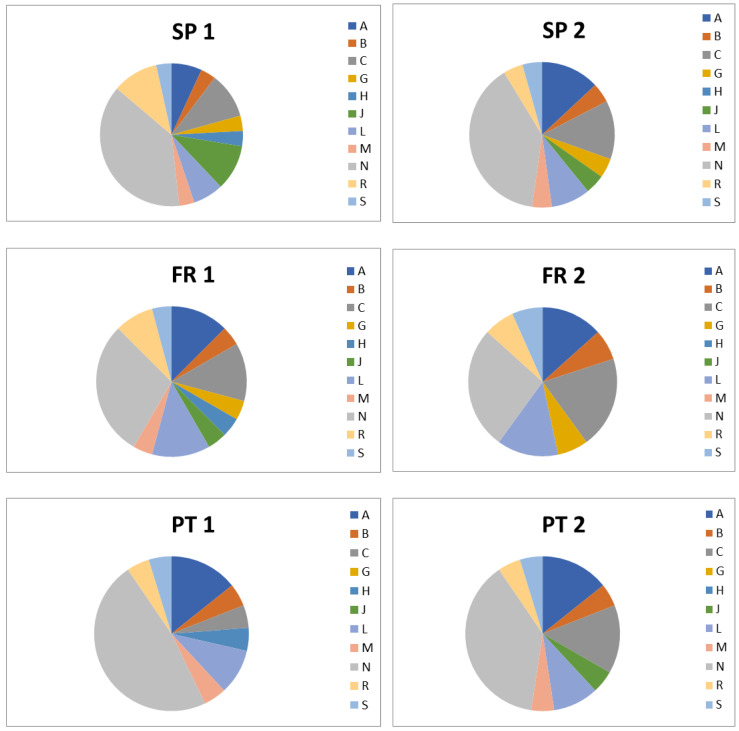
Profile of pharmaceuticals detected in each residence according to the ATC code: A = Alimentary tract and metabolism; B = Blood and blood forming organs; C = Cardiovascular system; G = Genito urinary system and sex hormones; H = Systemic hormonal preparations, excl. Sex hormones and insulins; J = Antiinfectives for systemic use; L = Antineoplastic and immunomodulating agents; M = Musculo-skeletal system, others; N = Nervous system; R = Respiratory system; S = Sensory organs.

**Figure 4 molecules-26-05047-f004:**
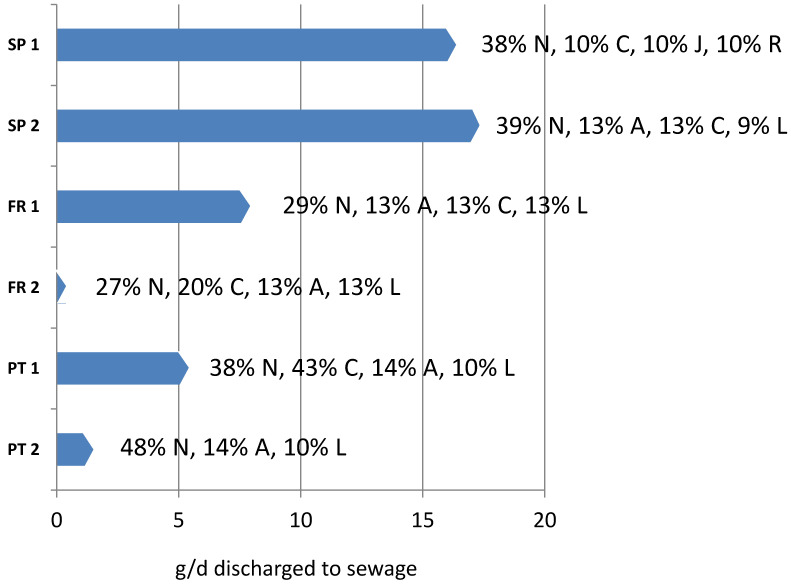
Total discharge of pharmaceuticals (g day^−1^) in the 6 senior residences and main families of compounds discharged according to ATC general category neurological drugs (N), cardiovascular (C), alimentary tract (A), neoplastic (L), anti-infective (J), respiratory (R).

**Table 1 molecules-26-05047-t001:** Characteristics of the residences sampled, indicating their origin, the capacity as number of beds, the type of residence, the water consumption, the number of pharmaceuticals administered during the period of sampling and the main compounds administered.

Residences	Capacity (Beds)	Type of Residence	Water Consumption (m^3^ year^−1^)	Number of Pharmaceuticals Administered	Number of Samples Collected
S1	130	Sociosanitary, general impairment & psychiatric unit	6679	164	15
S2	126	Housing and general impairment	7100	134	9
F1	81	Housing and general impairment	4560	133	15
F2	82	Housing and general impairment	5164	188	12
P1	52	Housing and general impairment	5230	116	8
P2	61	Housing and general impairment	4859	146	8

**Table 2 molecules-26-05047-t002:** Pharmacological properties of drugs used to determine their elimination and main metabolites excreted from each drug that can be potentially detected in water. Source: Drugbank [14].

Drug	Uses	Metabolism	Half-Life (h)	Elimination Route	Excreted Unchanged	Metabolites Formed
Dichlorobenzyl alcohol R02AA03	Mild antiseptic with a broad spectrum for bacterial and virus to treat mouth and throat infections	Hepatic		Urine	90%	hippuric acid
Amoxicillin J01CA04	Penicillin derivative used for the treatment of infections caused by gram-positive bacteria	Hepatic	1	Urine	70–78%	7 metabolites formed by hydroxylation; oxidative deamination; decarboxylation and glucuronidation
Amylmetacresol	Antiseptic used to treat infections in the mouth and throat.	n.a.	n.a.	Urine	n.a.	n.a.
Acetylsalicylic acid B01AC06	Salicylate used to treat pain, fever, inflammation, migraines, and reducing the risk of major adverse cardiovascular events	Hydrolyzed in plasma	0.2–0.3	Urine	10–85% as salicylate	salicylic acid; salicyluric acid; ether; phenolic glucuronide and ester; acyl glucuronide; gentisic acid; hydroxybenzoic acids
Atorvastatin C10AA05	Lipid-lowering drug to treat several types of dyslipidemias	P450 (rapid)	14	Bile	n.a.	para-hydroxyatorvastatin; para-hydroxyatorvastatin glucuronide; para and ortho-hydroxyatorvastatin lactone; ortho-hydroxyatorvastatin; ortho-hydroxyatorvastatin glucuronide
Bicalutamide L02BB03	Androgen receptor inhibitor used to treat metastatic prostate carcinoma.	Stereo specific metabolism	142	n.a.	n.a.	bicalutamide glucuronides
Budesonide D07AC09, R01AD05, A07EA06	Corticosteroid used to treat Crohn’s disease, asthma, COPD, hay fever and allergies, and ulcerative colitis.	CYP3A (>80-metabolized)	2–6	Urine	0% (60% as metabolite)	6-beta-hydroxybudesonide; 22-hydroxy Intermediate; 16alpha-butyrloxyprednisolone; 16-alpha-hydroxyprednisolone; delta6-budesonide; 23-hydroxybudesonide
Caffeine N06BC01	Stimulant present in tea, coffee, cola beverages, analgesic drugs, and agents used to increase alertness.	Hepatic CYP1A2	5	Urine	2% (mainly adsorbed in renal tubules)	theobromine; theophylline; 1-methylxanthine; 1-methyluric acid; 1,3,7-trimethyluric acid; 1,7-dimethylxanthine; 1-methylxanthine; 5-acetylamino-6-formylamino-3-methyluracil; 1,7-dimethyluric acid
Carbamazepine N03AF01	Anticonvulsant to treat seizures and pain resulting from trigeminal neuralgia	Hepatic CYP3A4	27–36.8	Urine	72%	10,11-epoxycarbamazepine; 10,11-dihydroxycarbamazepine; 3-hydroxycarbamazepine; 2,3-dihydroxycarbamazepine; carbamazepine-*o*-quinione; carbamazepine 2,3-epoxide; 2-hydroxycarbamazepine; 2-hydroxyiminostilbene; iminoquinone
Clomethiazole N05CM02	γ-aminobutyric acid (GABAA)-mimetic drug used as sedative and hypnotic to prevent symptoms of acute alcohol withdrawal.	n.a.	n.a.	n.a.	n.a.	n.a.
Chlorpheniramine R06AB04	Histamine to treat upper respiratory allergies	Hepatic CYP 450	21–27	n.a.	n.a.	n.a.
Clarythromycin J01FA09	Macrolide antibiotic	Hepatic-CYP3A4	3–4	Urine	20–30%	14-hydroxyclarithromycin; *N*-desmethylclarithromycin
Cloperastine R05DB21	Cough suppressant	n.a.	n.a.	n.a.	n.a.	n.a.
Cyclophosphamide L01AA01	Nitrogen mustard to treat lymphomas, myelomas, leukemia, mycosis fungoides, neuroblastoma, ovarian adenocarcinoma, retinoblastoma, and breast carcinoma.	Hepatic CYP450 (75% of the drug activated)	3–12	Urine	10–20%	4-hydroxycyclophosphamide; aldophosphamide; phosphoramide mustard; phosphoramide aziridinium; acrolein; acrylic acid; carboxyphosphamide; alcophosphamide; carboxyphosphamide; nornitrogen mustard; 4-ketocyclophosphamide; dechloroethyl cyclophosphamide; chloroacetaldehyde
Dextromethorphan R05DA09	To treat cases of dry cough	CYP3A4, CYP2D6, CYP2C9; CYP2D6 CYP2C9	3–30	n.a.	n.a.	dextrorphan; 3-hydroxymorphinan; 3-hydroxymorphinan sulfate; 3-hydroxymorphinan *o*-glucuronide; dextrorphan sulfate; dextrorphan *o*-glucuronide; (+)-3-methoxymorphinan 3-hydroxymorphinan
Diclofenac M01AB05, others	SAID used to treat pain and inflammation	CYP2C9	2	Urine	60–70%	4’-hydroxydiclofenac; diclofenac acyl glucuronide; 3’-hydroxydiclofenac; 5-hydroxy diclofenac; diclofenac *o*-imine methine; diclofenac 2’;3’-oxide; diclofenac radical and the hydroxylated and gluthathion conjugated derivatives
Donepezil N06DA02	Acetylcholinesterase inhibitor to treat mild to moderate Alzheimer’s Disease	Hepatic CYP3A4	70	Urine	57%	6-*O*-desmethyl donepezil and *O*-dealkylation, hydroxylation, *N*-oxidation, hydrolysis, and *O*-glucuronidation metabolites
Dutasteride G04CB02	Antiandrogenic compound to treat benign prostatic hyperplasia	Hepatic CYP3A4, CYP3A5	840	Feces	1–15% (2–90% as metabolites)	4′-hydroxydutasteride; 6-hydroxydutasteride; 6,4′-dihydroxydutasteride; 1,2-dihydrodutasteride; 15-hydroxydutasteride; 6,4′-dihydroxydutasteride; 15-hydroxydutasteride
Escitalopram N06AB10	Serotonin re-uptake inhibitor to treat depressive, anxiety and obsessive-compulsive disorders	Hepatic, CYP2C19, CYP3A4 CYP2D6	27–32	Urine	8%	*S*-desmethylcitalopram; *S*-didesmethylcitalopram; escitalopram propionic acid
Estrone G03CA07	Estrogen to treat perimenopausal and postmenopausal symptoms	Hepatic	19	n.a.	n.a.	2-hydroxyestrone; 4-hydroxyestrone; estrone sulfate; estrone glucuronide; 2-hydroxyestrone sulfate; 4-hydroxyestrone sulfate; 2-OH-estrone; 4-OH-estrone; 6alpha-*H*-estrone; 6beta-estrone; 7alpha-OH-estrone; 15alpha-OH -estrone; 16alpha-OH-estrone; 16beta-OH-estrone
Ezetimibe C10AX09	Cholesterol absorption inhibitor used to lower total cholesterol	Small intestine and hepatic	22	Feces	69%	ezetimibe glucuronide; SCH 57871; SCH 57871-glucuronide, SCH 488128
Fluticasone R03BA05	Corticosteroid to treat and manage asthma, dermatoses	Hepatic CYP 450 3A4	24	Feces	90%	fluticasone propionate
Furosemide C03CA01	Diuretic to treat hypertension, and edemas	Kidney Hepatic	4.5	Urine	50%	furosemide glucuronide; saluamine; 4-chloro-5-sulfamoylanthranilic acid
Gabapentin N03AX12	Anticonvulsant for peripheral neuropathy, neuralgia and seizures	Not metabolized	5–7	Urine	100%	No metabolites; excreted as unchanged drug
Ibuprofen M02AA13, others	NSAID analgesic, anti-inflammatory and antipyretic	Hepatic (rapid biotransformed)	1.2–2	Urine	0%	ibuprofen glucuronide; 2-hydroxyibuprofen; 3-hydroxyibuprofen; carboxy-ibuprofen; 1-hydroxyibuprofen
Ifosfamide L01AA06	Alkylating and immunosuppressive agent used in chemotherapy for the treatment of cancers	Hepatic	7–15	Urine	12–18%	3-dechloroethylifosfamide; dechloroethylifosfamide; chloroacetaldehyde; 4-hydroxyifosfamide and derivatives
L-ascorbic acid A11GA01	Vitamin used to correct vitamin C deficiency, scurvy, iron sorption	Hepatic	384	n.a.	n.a.	ascorbic acid-2-sulfate; dehydroascorbate; 2;3-dikeogulonic acid; erythrulose; threosone; oxalic acid
Levetiracetam N03AX14	Novel anticonvulsant agent to treat onset seizures in epileptic patients	Minimally metabolized	6–8	Urine	66%	levetiracetam carboxilic acid
Levofloxacin J01MA12	Fluoroquinolone antibiotic to treat infections	n.a.	6–8	Urine	87%	desmethyl-levofloxacin levofloxacin-*N*-oxide
Lidocaine S01HA07, D04AB01, R02AD02, C01BB01, N01BB52	Local anesthetic used in superficial and invasive procedures	Hepatic (rapid)	1.3–2	Urine	5%	3-hydroxylidocaine; monoethylglycinexylidide; 2,6-dimethylaniline (2,6-xylidine)and hydroxyl and amino derivatives
Macrogol A06AD15	Laxative to treat constipation and used before colonoscopies	Not metabolized	4.1	Feces	85–99%	No metabolites formed
Megestrol G03AC05, others	Progestin to treat anorexia and cachexia and antineoplastic agent	Hepatic	34	Urine	n.a.	n.a.
Mycophenolic acid L04AA06	Immunosuppressant to prevent organ transplant rejections	Glucuronyl transferase	8–16	n.a.	n.a.	mycophenolic acid-acyl glucuronide; 6-O-desmethyl-mycophenolic acid; mycophenolic acid-7-O-glucuornide
Paracetamol N02BE01	Analgesic drug for pain management and as antipyretic	Hepatic	2.5	Urine	5%	*N*-acetyl-*p*-benzoquinone imine; acetaminophen cysteine, glucuronide and sulfate
Prednisone H02AB07	Corticosteroid to treat inflammation, immune-mediated reactions and endocrine or neoplastic diseases	n.a.	2–3	Urine	n.a.	prednisolone and hydroxy and dihydro prednisone and further metabolized to sulfate and glucuronide conjugates
Pregabalin N03AX16	Anticonvulsant drug to treat neuropathy, fibromyalgia, seizures	Less than 2%	6.3	Urine	98%	*N*-methylpregabalin
Quetiapine N05AH04	Psychotropic for the management of bipolar disorder, schizophrenia, and major depressive disorder	Hepatic	6–7	Urine	1%	*n*-desalkylquetiapine; 7-hydroxyquetiapine; quetiapine sulfoxide; *o*-desalkylquetiapine
Rosuvastatin C10AA07	Lipid regulator	Hepatic	19	Feces	77%	*n*-desmethylrosuvastatin; rosuvastatin 5 S-lactone
Sulfamethoxazole J01EC01	Sulfonamide antibiotic to treat urinary, gastrointestinal and respiratory infections	Kidney	10	Urine	30%	5-hydroxysulfamethoxazole; *N*-acetylsulfamethoxazole; sulfamethoxazole N4-hydroxylamine; sulfamethoxazole *N*-glucuronide
Tiotropium R03BB04	Bronchodilator to treat chronic obstructive pulmonary disease	Not heavily metabolized	24	Urine	74%	*N*-methylscopine + dithienylglycolic acid
Trazodone N06AX05	Serotonin uptake inhibitor to treat major depressive disorder	Hepatic CYP3A4	7.3	Feces	21%	*m*-chlorophenylpiperazine; triazolopyridinone dihydrodiol; triazolopyridinone epoxide; 4’-hydroxytrazodone
Valsartan C09CA03	Angiotensin-receptor blocker to manage hypertension	Minimal hepatic	6	Feces	83%	valeryl-4-hydroxyvalsartan
Vildagliptin A10BH02	Treatment of type 2 diabetes mellitus	Hydrolyzed	2	Urine	23%	vildagliptin M20.7; M15.3; M20.2; M20.9;M21.6 metabolites

## Data Availability

The data presented in this study are available on request from the corresponding author.

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
