# Peer review of "Pharmaceutical Residues in Senior Residences Wastewaters: High Loads, Emerging Risks"

_molecules, 2021, doi:10.3390/molecules26165047_

Round 1

Reviewer 1 Report

The article is well-written, and it can be published after revision.

  1. Statistical analysis data are missing and should be included in tables and additional Figures.
  2. Figures and tables should be placed near the text describing them.
  3. Figure 1. SP, PT and FR , probably Spain, Portugal and France should be explained in the Figure legend How do you explain the great difference of total pharmaceuticals detected in water at specific days of the week? morning and after- noon flush waters samples should be separated in this Figures for comparison purposes among different countries.
  4. Table 1 and Table 2 could go to the supplement
  5. The authors mention that : “We found some variability on the detected concentrations according to the sampling period suggesting that the punctual sampling provides a “picture” of the precise moment that bathrooms are flushing.” This variability should be presented in Figures.
  6. The intra-day and inter-day variability was high in SP and PT…. Data on the statistical analysis of the analyzed samples from France, Spain and Portugal are missing and should be included in a table. Along with statistical analysis of the samples between different time intervals.

Author Response

The article is well-written, and it can be published after revision.

1. Statistical analysis data are missing and should be included in tables and additional Figures.

Reply: Thanks for the comment. Statistical analysis has been included in the new version to show differences among residences and correlations among sampling days and times.

2. Figures and tables should be placed near the text describing them.

Reply: We placed the figures and tables into the main text close to their first citation as requested, thank you.

3. Figure 1. SP, PT and FR , probably Spain, Portugal and France should be explained in the Figure legend. How do you explain the great difference of total pharmaceuticals detected in water at specific days of the week? morning and after- noon flush waters samples should be separated in this Figures for comparison purposes among different countries.

Reply: Spain, France and Portugal have been indicated in the legend, thank you. The great differences of total pharmaceuticals detected in the different days (and hours) of the week largely vary because grab sampling is only able to detect those contaminants in the punctual period that sampling has been performed, so there is high variability depending to “the chance” to collect the flushes with the highest levels of pharmaceuticals. This has been indicated in the new version. Morning and afternoon flushes were already separated in Figure 1 to indicate that the total loads of pharmaceuticals discharged by the different senior residences vary according to the time sampled and that these variations can be high. This has been clarified in the legend.

4. Table 1 and Table 2 could go to the supplement

Reply: Regarding Table 1, in the new version of the manuscript we have merged table 1 and 3 as this way it is easier for the reader to see the compounds studied that were selected according to their consumption by the elderly and their uses, which help the reader identify what are their main prescriptions. In addition, information on the metabolization potential permit to verify the concentrations detected in water. Regarding Table 2, we prefer to maintain this information on the body of the paper as is provides the specific information on the waste waters studied from each residence.

5. The authors mention that : “We found some variability on the detected concentrations according to the sampling period suggesting that the punctual sampling provides a “picture” of the precise moment that bathrooms are flushing.” This variability should be presented in Figures.

Reply: The intra-day and inter-day variability observed in the concentrations detected are indicated in Figure 1 considering the total concentration of pharmaceuticals. Figure 2 shows the levels considering all samplings in each residence for each compound and is true that the daily variability cannot be appreciated, but it was difficult to show this variability due to the large number of compounds detected. In the new version we have provided a table with the concentrations detected at each sampling period for each individual compound as supporting information.

6. The intra-day and inter-day variability was high in SP and PT…. Data on the statistical analysis of the analyzed samples from France, Spain and Portugal are missing and should be included in a table. Along with statistical analysis of the samples between different time intervals.

Reply: Statistical analysis has been included in the new version.

Reviewer 2 Report

The authors studied the presence of 43 pharmaceuticals highly consumed by the elderly population in 6 senior residences located in France, Portugal and Spain. The final aim is to highlight the need to implement at-source waste water treatment procedures in senior residences, which have been identified as a point source pollution of pharmaceuticals. The research is of great significance and the results have referential value. I recommend publishing this work after minor revision:

  1. Given the large amount of non-original information in Tables 1 and 3, please explain how does each of these statistics contribute to research.
  2. According to the data in Table 2, please explain what factors determine your sample size.
  3. How are so many and complex pharmaceuticals determined by liquid chromatography, and what are the analytical errors involved?

Author Response

Referee 2. The authors studied the presence of 43 pharmaceuticals highly consumed by the elderly population in 6 senior residences located in France, Portugal and Spain. The final aim is to highlight the need to implement at-source waste water treatment procedures in senior residences, which have been identified as a point source pollution of pharmaceuticals. The research is of great significance and the results have referential value. I recommend publishing this work after minor revision:

1. Given the large amount of non-original information in Tables 1 and 3, please explain how does each of these statistics contribute to research.

Reply: The referee is right and this information in Tables 1 and 3 is extracted for the Drugbank database to provide information on the compounds tested and was included to support our findings. As these are 2 large tables, in the new version of the manuscript we have merged to one so that the reader can better identify the compounds studied, their uses by the elderly, their metabolisation and metabolites formed. Altogether, this information is key to understand the high concentrations detected in waters from senior residences and provides also information on main metabolites that can be new future contaminants to be surveyed.

2. According to the data in Table 2, please explain what factors determine your sample size.

Reply: We have inserted “The number of samples collected in each residence are listed in Table 2 and varied depending on the accessibility in each residence and availability of maintenance personnel that was essential to obtain water from the chests which were located inside the residence’s premises”.

3. How are so many and complex pharmaceuticals determined by liquid chromatography, and what are the analytical errors involved?

Reply: The method used was developed and validated in a previous study (Ref. 13. Gómez-Canela, C., Sala-Comorera, T., Pueyo, V., Barata, C., Lacorte, S. Analysis of 44 pharmaceuticals consumed by elderly using liquid chromatography coupled to tandem mass spectrometry. J. Pharmac. Biomed. Anal. 2019, 168, 55) that aimed to optimize the extraction conditions in an effective way to obtain high recovery yields with small errors. In addition, the chromatographic and mass spectrometry conditions were also optimized to identify multi-compounds, and this was important because it permitted to detect main pharmaceuticals in water from senior residences according to consumption patterns (Ref. 14. Gómez-Canela, C., Pueyo, V., Barata, C., Lacorte, S., Marcé, R.M. Development of predicted environmental concentrations to prioritize the occurrence of pharmaceuticals in rivers from Catalonia. Sci. Total Environ. 2019, 666, 57). The analytical errors of the method developed are

Round 2

Reviewer 1 Report

The authors have greatly improved the manuscript and replied to all the comments raised by the reviewers. I would suggest publication in the present form